

# Continuous training in young athletes decreases hepcidin secretion and is positively correlated with serum 25(OH)D and ferritin

Yukiko Kobayashi, Rikako Taniguchi, Emiko Shirasaki, Yuko Segawa Yoshimoto, Wataru Aoi and Masashi Kuwahata

Graduate School of Life and Environmental Sciences, Kyoto Prefectural University, Kyoto, Japan

## ABSTRACT

**Background:** Iron deficiency is known to impair muscle function and reduce athletic performance, while vitamin D has been reported to induce iron deficiency. However, the mechanism underlying exercise-induced changes in iron metabolism and the involvement of vitamins in this mechanism are unclear. The present study examined changes in biological iron metabolism induced by continuous training and the effects of vitamin D on these changes.

**Methods:** Diet, physical characteristics, and blood test data were collected from 23 female high school students in a dance club on the last day of each of a 2-month continuous training period and a 2-week complete rest periods.

**Results:** Serum hepcidin-25 levels were significantly lower during the training period than the rest period ($p = 0.013$), as were the red blood cell count, hemoglobin, and hematocrit (all $p < 0.001$). Serum erythropoietin was significantly higher ($p = 0.001$) during the training period. Significant positive correlations were observed between 25(OH)D levels and serum iron, serum ferritin, and transferrin saturation during the training period. Multiple regression analysis with serum 25(OH)D level as the dependent variable and serum ferritin and iron levels as independent variables during the training period revealed a significant association with serum ferritin.

**Conclusion:** Continuous training may promote hemolysis and erythropoiesis, contributing to the suppression of hepcidin expression. The relationship between serum 25(OH)D and iron *in vivo* may be closely related to metabolic changes induced by the exercise load.

## INTRODUCTION

Anemia is a condition in which the amount of hemoglobin in the circulating blood decreases and the oxygen-carrying capacity of the blood is reduced, resulting in insufficient oxygen supply to the tissues throughout the body. Moreover, iron deficiency, with or without anemia, can impair muscle function, limit work capacity, and reduce adaptation to training and athletic performance (*Giovanni, Giuseppe & Giuseppe, 2014*).

Corresponding author
Yukiko Kobayashi,
yukicoba@kpu.ac.jp

Exercise-induced anemia (sports anemia) results from iron deficiency due to increased iron efflux and demand, including exercise-induced hemolysis, increased iron loss through sweat and urine, and increased skeletal muscle growth (*Siegel et al., 1979*; *Stewart et al., 1984*; *Sinclair & Hinton, 2005*). The prevalence of iron deficiency in athletes is higher than that in the general population. Iron deficiency is reported to be 25–35% in adolescent and adult women and 11–36% in men, with a wide range depending on the sport (*Yamamoto et al., 2022*). Further, a survey of 13–22-year-old participants in school sports clubs in Japan showed anemia in 16.5% of the participants (*Andrews, 2008*). Therefore, it is not difficult to imagine that many athletes, regardless of age, sex, sports discipline, or level of competition, have sports anemia.

The mechanisms of iron metabolism have been rapidly elucidated in recent years, and the hepcidin-ferroportin system is understood to be the central regulatory mechanism of iron metabolism *in vivo*. Ferroportin (FPN) is a membrane protein that pumps iron out of cells, and hepcidin binds to FPN, which then internalizes and degrades the FPN export channels. Decreased FPN in intestinal epithelial cells and macrophages due to increased hepcidin levels inhibits iron absorption in the gastrointestinal tract and iron release from macrophages, and the amount of transferrin-bound iron available *in vivo* is regulated. Therefore, hepcidin serves as an *in vivo* iron regulator (*Hentze et al., 2010*; *Ganz & Nemeth, 2012*).

Hepcidin secretion is thought to be regulated by various factors, including iron saturation signals *via* bone morphogenetic protein (BMP) and inflammatory signals *via* interleukin-6 (IL-6), which enhance hepcidin expression, whereas erythropoietic and hypoxic signals suppress hepcidin expression (*Peeling, 2010*). Increased inflammatory signaling was observed after exercise. For example, many reports show that IL-6 blood levels peak immediately after exercise, and muscle damage from prolonged exercise causes a sustained increase in IL-6, resulting in increased hepcidin secretion (*Latunde-Dada, 2013*; *Holick, 2007*). Hepcidin secretion may also be affected by signals of erythropoiesis in response to exercise-induced hemolysis and hypoxia due to organ ischemia. Thus, a consensus is emerging on exercise-induced changes in iron metabolism, but the mechanisms are still under debate.

Various relationships between iron metabolism and fat-soluble vitamins, particularly vitamin D, have recently been reported. Vitamin D is a fat-soluble prohormone that plays an important role in the endocrine and autocrine processes *in vivo*, mainly by maintaining adequate blood calcium and phosphate levels. Vitamin D is also involved in the body's inflammatory response through the activation and differentiation of immune and inflammatory cells and has recently been the subject of extensive research on muscle mass, strength, and function (*Rosendahl-Riise et al., 2017*; *Bacchetta et al., 2014*). Furthermore, it has been reported that vitamin D is associated with the regulation of iron metabolism through its effect on the iron regulator hepcidin (*Smith & Tangpricha, 2015*), suggesting that reduced vitamin D levels may consequently induce iron deficiency and anemia (*Mogire et al., 2022*). In contrast, a study on the relationship between vitamin D and iron in humans reported a high frequency of iron deficiency in various life stages, including

vitamin D-deficient children, the elderly, young adults, and athletes (*Sim et al., 2010*; *Malczewska-Lenczowska et al., 2018*).

Although iron deficiency is likely to occur during exercise due to increased iron efflux and demand, there is limited research on hepcidin secretion as an influencing factor, particularly with regard to the effects of prolonged and continuous training. Furthermore, few reports have examined the relationship between vitamin D, which has been suggested to have a direct effect on hepcidin secretion and is involved in muscle and bone formation, and exercise-induced iron metabolism dynamics. It would be of great interest to know if there are specific iron metabolic mechanisms that are produced by continuous exercise loading, and if dietary vitamin D has an effect on these mechanisms. The aim of the present study was therefore to observe the *in vivo* iron metabolism dynamics of exercise with a focus on hepcidin in young athletes undergoing prolonged and continuous training, using periods of training inactivity in the same participants group as a control. Furthermore, the influence of vitamin D on the *in vivo* iron status during training was investigated from both dietary and body concentration aspects.

## MATERIALS AND METHODS

Portions of this text were previously published as part of a preprint (*Kobayashi et al., 2024*).

### Ethical issues, including informed consent

This study was conducted with the approval of the Kyoto Prefectural University Ethics Committee (No. 207, Date approved: August 4 2020). Participants were asked orally and in writing to participate in the study, and written informed consent was obtained from the participants. Informed consent was also obtained from the guardians of participants.

### Participants

Twenty-six high school students from a competitive dance club were included in the analysis; of these, 23 were female (mean age 16.4 ± 0.5 years). These 23 female were targeted for analysis.

### Data collection

The study was divided into two phases: a training phase (6.3 Mets/3 h of training/day, 7 days a week, approximately 2 months passed) and a rest phase (test period without extracurricular activities, approximately 2 weeks passed without training). Participants' physical characteristics, and blood biochemical tests were assessed on the last day of each of a 2-month continuous training period and a 2-week complete rest periods. Participants' dietary survey and energy consumption were recorded on the last 1 week of each of a training period and a rest period. The survey was conducted in 2020 for both the training and rest periods.

For the dietary survey, participants were asked to record the contents of their meals for 2 non-consecutive days and capture photographs (meal and measurement together) with a digital camera. From the collected recording forms and photographs, the daily intake of energy and nutrients (energy, protein, fat, carbohydrate, iron, and vitamin D) was

calculated using dedicated nutrition calculation software (Excel Eiyoukun Ver. 9, Kenpakusha, Tokyo).

For 2 days, on the same day as the dietary record, the participants were asked to record their activities at 5-min intervals from the time of waking to the time of going to bed. Exercise intensity and duration were calculated from activity records, referring to the Japanese version of physical activity codes and MET values in the "2011 Compendium" (*Ainsworth et al., 2011*). The exercise intensity was measured using an activity meter (HJA-750C Active Style Pro; Omron, Kyoto, Japan). The participant's daily energy expenditure was calculated using the following formula; daily energy expenditure (kcal/day) = body weight $\times$ 1.05 $\times$ $\Sigma$ (exercise intensity METs $\times$ time).

Height was measured using a stadiometer, and body weight, lean body mass, skeletal muscle mass, and body fat mass were measured using a impedance body composition analyzer (Inbody 270, Inbody Japan, Tokyo, Japan).

Blood samples were drawn once during the training period and once during the rest period. During the training period, blood samples were collected in the morning before the start of training. During the rest period, blood samples were taken in the afternoon after regular examinations in the morning and eating lunch. Blood properties and biochemical tests, such as red blood cell (RBC) count, hemoglobin (Hb) level, hematocrit (Ht), reticulocyte count, and serum erythropoietin (EPO) level, were performed in a laboratory (Kyoto Microbio Laboratory, Kyoto, Japan). The RBC count and Ht were measured by sheath flow DC detection, Hb by the SLS hemoglobin method, reticulocytes by flow cytometry, and EPO by CLEIA. Serum hepcidin-25, iron levels, and unsaturated iron-binding capacity (UIBC) measurements, were commissioned (MC Plot Biotechnology, Kanazawa, Japan) using several methods. Serum hepcidin-25 levels were measured by liquid chromatography with a tandem mass spectrometry assay system (4000 QTRAP; Applied Biosystems, Waltham, MA, USA). Liquid chromatography with tandem mass spectrometry assay system comprised LC (Prominence LC20-ADvp, Shimadzu, Kyoto, Japan); 2.1 $\times$ 150 mm, ZORBAX 300 SB-C8 column (Agilent Technologies, Waltham, CA, USA), and flow velocity of 300 mL/min. Serum iron and UIBC were measured using the nitroso PSAP method. Total iron-binding capacity (TIBC) and serum transferrin saturation were calculated as follows: TIBC = serum iron + UIBC; serum transferrin saturation = serum iron/TIBC$\times$100. Serum IL-6 levels were measured using an enzyme-linked immunosorbent assay kit (KE10007; Protein Tech Japan, Tokyo, Japan). Serum 25-hydroxyvitamin D (25(OH)D) levels were measured using enzyme-linked immunosorbent assay kits (25-OH Vitamin D Total ELISA Kit, KA6138, Abnova, Taipei, Taiwan).

## Statistical analyses

The data were presented as mean $\pm$ standard deviation. The Shapiro–Wilk test was used to test for normality. For comparisons between the training and rest periods, a paired t-test was used for normally distributed data and a Wilcoxon signed-rank sum test for non-normally distributed data. To examine the correlation between serum 25(OH)D and blood cell parameters, iron metabolism-related parameters, physical characteristics, and

dietary factors, Spearman's rank correlation coefficient test was used with serum 25(OH)D as the dependent variable. To investigate iron metabolism-related parameters affecting serum 25(OH)D during the training phase, ferritin, serum iron and transferrin saturation, which showed significant single correlations, were selected as candidate independent variables. Preliminary observations of the correlation between serum iron, ferritin and transferrin saturation showed that the correlation between serum iron and transferrin saturation was r > 0.8 and the correlation between serum iron and ferritin was r < 0.8, so serum iron and ferritin were finally selected as independent variables in view of multicollinearity issues. Multiple regression analysis was performed using the forced entry method with serum iron and ferritin as bivariate independent variables and serum 25(OH)D as the dependent variable. The variance inflation factor was less than 10 and there were no multicollinearity issues. IBM SPSS Statistics 25 (Armonk, NY, USA) was used for all of the above statistical analyses. The significance level for all analyses was 5%.

Sample size calculation and *post hoc* power analysis were performed using G *Power 3.1.9.6 (*Faul et al., 2007*). Assuming a two-tailed paired t-test with an effect size of 0.77, a significance level of 5%, and a power of 80%, the required sample size was calculated to be 16 cases. An effect size was estimated using pre- and post-exercise serum hepcidin levels in previous studies. The power was greater than 80% in a *post hoc* power analysis (1-$\beta$ = 0.883). In addition, based on the results of the correlation analysis between serum ferritin and 25(OH)D in the training group, assuming a two-tailed test for the population correlation coefficient with an effect size of 0.52 and a significance level of 5%, the power was calculated to be 79.5% in a *post hoc* power analysis.

## RESULTS

### Physical characteristics of the participants

The physical characteristics of the participants are shown in Table 1. The percentage of participants with a BMI less than 18.5 was 35% in the training phase and 26% in the rest phase. BMI ($p = 0.009$) and body fat mass ($p < 0.001$) were significantly lower during the training phase than during the rest phase, whereas lean body mass and skeletal muscle mass were significantly higher (both $p < 0.001$).

### Energy and nutrients intake

Table 1 shows the nutrient and other intakes of the participants. There was no significant difference in energy intake between the two groups; however, intake/consumption was significantly lower during the training period than during the rest period ($p < 0.001$), with 87% of the participants having an energy intake less than their consumption. Carbohydrate intake was higher during the training period ($p = 0.057$) and fat intake was significantly lower ($p < 0.05$) than during the rest period. Vitamin D and iron intakes were not significantly different between the two periods.

### Blood and biochemical test data

The participants' blood properties and biochemical data of the participants are shown in Table 1 and Fig. 1. The erythrocyte count, hemoglobin concentration, and hematocrit

![PeerJ]

**Table 1 The physical characteristics, the nutrient and other intakes, the blood properties and biochemical data of the participants during training and resting phase.**

|  | Training<br>$n = 23$ | Resting<br>$n = 23$ | $p$ value |
|---|---|---|---|
| BMI (kg/m$^2$) | 19.5 ± 1.9 | 19.8 ± 2.0 | 0.009* |
| <18.5 (%) | 34.8 | 26.1 | – |
| Lean body mass (kg) | 38.9 ± 3.6 | 38.3 ± 3.6 | <0.001* |
| Skeletal muscle (kg) | 21.0 ± 2.1 | 20.8 ± 2.1 | <0.001* |
| Fat mass (kg) | 9.6 ± 3.0 | 10.8 ± 3.1 | <0.001* |
| Energy consumption (kcal) | 2,432 ± 295 | 1,863 ± 255 | <0.001* |
| Energy intake (kcal) | 1,990 (1,546–2,116) | 1,880 (1,633–2,248) | 0.301** |
| Energy intake/consumption (%) | 77.3 ± 17.5 | 105.4 ± 28.8 | <0.001* |
| Dietary protein (g) | 69.5 ± 19.3 | 71.8 ± 19.2 | 0.629* |
| Dietary protein/BW (g/kg BW) | 1.5 ± 0.5 | 1.5 ± 0.5 | 0.714* |
| Dietary fat (E%) | 31.5 ± 5.9 | 33.6 ± 7.6 | <0.001* |
| Dietary carbohydrate (E%) | 53.2 (47.2–56.8) | 46.9 (44.0–50.9) | 0.057** |
| Dietary vitamin D (mg) | 3.9 (1.8–7.1) | 6.9 (6.0–10.2) | 0.884** |
| Dietary iron (mg) | 7.0 (6.7–8.9) | 3.3 (2.4–7.6) | 0.465** |
| Iron deficiency anemia[†] (%) | 8.7 | 0 | – |
| Red Blood Cell count (×10$^6$/μl) | 434 ± 25 | 460 ± 23 | <0.001* |
| Hemoglobin (g/dl) | 12.7 ± 0.7 | 13.3 ± 0.6 | <0.001* |
| <12 g/dl (%) | 21.7 | 0 | – |
| Hematocrit (%) | 38.7 ± 2.1 | 40.1 ± 1.9 | <0.001* |
| MCV (fl) | 88.6 ± 3.0 | 85.9 ± 3.4 | <0.001* |
| MCH (pg) | 29.2 ± 1.2 | 28.8 ± 1.3 | <0.001* |
| MCHC (%) | 32.7 ± 0.7 | 33.0 ± 0.6 | 0.013* |
| Reticulocytes (‰) | 13.6 ± 4.1 | 12.8 ± 4.7 | 0.235* |
| Serum iron (μg/dl) | 70 (50–111) | 94 (77–118) | 0.162** |
| Total iron-binding capacity (μg/dl) | 360 (337–385) | 391 (362–422) | 0.044** |
| Serum transferrin saturation (%) | 19.4 (13.5–29.6) | 24.8 (19.7–29.3) | 0.394** |
| Serum ferritin (ng/ml) | 20.7 (11.0–42.7) | 23.8 (11.4–41.4) | 0.903** |
| <12 ng/ml (%) | 26.1 | 26.1 | – |
| Serum hepcidin-25 (ng/dl) | 7.2 (1.6–12.2) | 16.8 (9.4–30.3) | 0.004** |
| Serum IL-6 (pg/ml) | 1.3 ± 0.1 (n = 3) | 1.2 ± 0.2 (n = 3) | – |
| Serum erythropoietin (mlU/ml) | 10.7 ± 3.5 | 8.0 ± 2.0 | 0.001* |
| Serum 25(OH)D (ng/ml) | 34.5 ± 6.6 | 33.3 ± 7.9 | 0.471* |

**Note:**
Value are mean ± SD or Median (IQR). *Paired t-test or **Wilcoxson signed-rank test. [†]Hb < 12 g/dl and Serum ferritin <12 ng/ml. BMI, body mass index; MCV, mean corpuscular volume; MCH, mean corpuscular hemoglobin; 25(OH)D, 25-hydroxyvitamin D.

values were significantly lower during the training period than during the rest period (all $p < 0.001$). The reticulocyte count was not significantly different between the two periods but was slightly higher during the training period, and the hematopoietic factor EPO was significantly higher during the training period than during the rest period ($p = 0.001$). Serum iron and transferrin saturation did not differ significantly between the two periods,

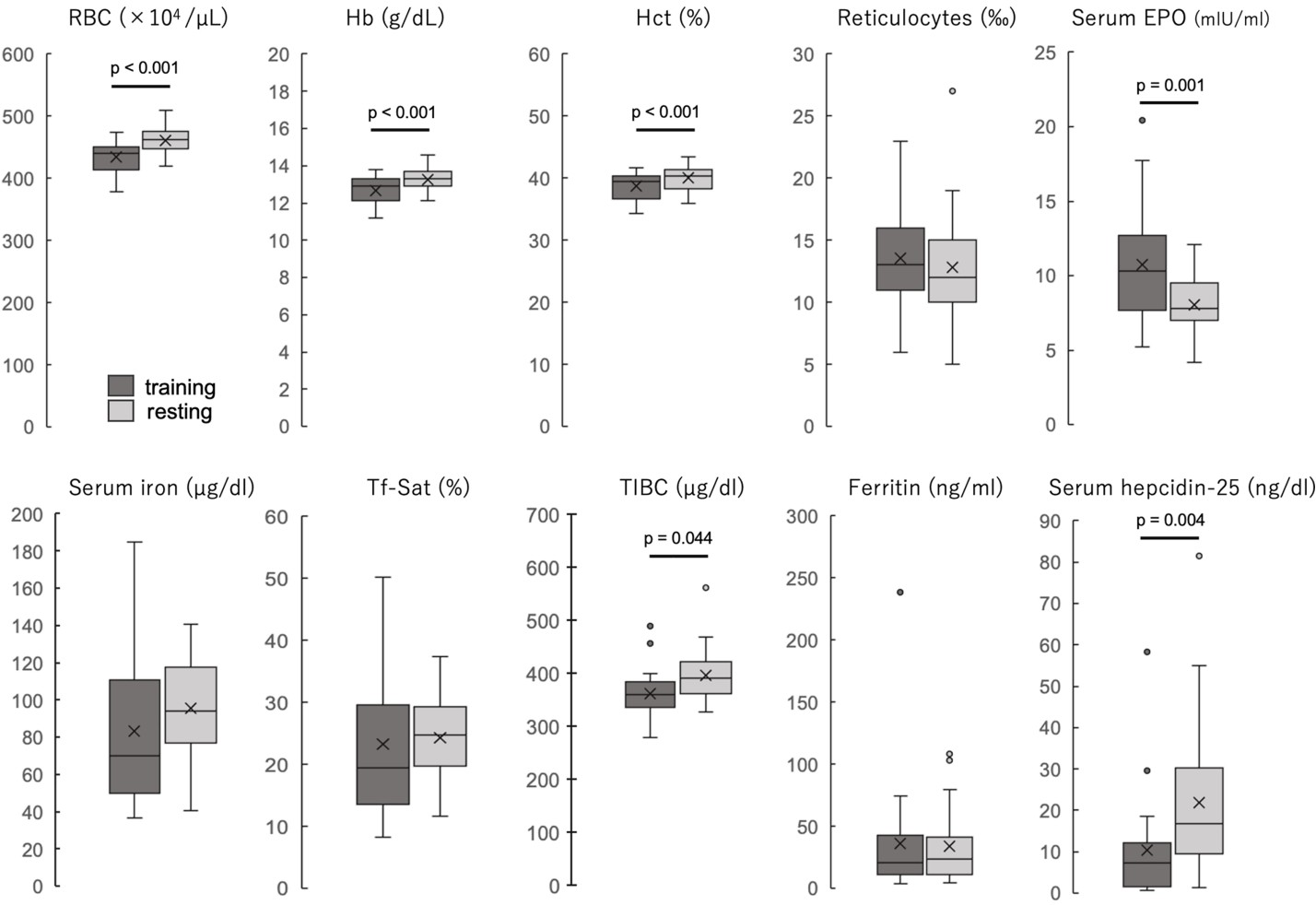

**Figure 1** **The blood properties and biochemical data of the participants during training and resting periods.** Dark gray boxes indicate training period; light gray boxes, resting period. All groups are *n* = 23. Paired t-test (RBC, Hb, Hct, reticulocytes, and serum EPO) or Wilcoxson signed-rank sum test (serum iron, Tf-Sat, TIBC, ferritin, and serum hepcidin-25). The center line of the boxplot indicates the median, the crosses indicate the mean, the top of the box indicates the 75th percentile, and the bottom of the box indicates the 25th percentile. RBC, red blood cell count; Hb, hemoglobin; Ht, hematocrit; EPO, erythropoietin; Tf-Sat, transferrin saturation; TIBC, total iron-binding capacity.

but the median values tended to be lower and varied more during the training period. TIBC was significantly lower during the training period than during the rest period. Ferritin levels were not significantly different between the two periods. Serum hepcidin-25 levels were significantly lower during the training period than those during the rest period (*p* = 0.004). Serum 25(OH)D levels were not significantly different between the two periods, and IL-6 was detected in only three participants in both groups, while it was below the detection limit in the remaining 20 participants. When iron-deficiency anemia was defined as a hemoglobin concentration less than 12 g/dl and serum ferritin less than 12 ng/dl, two (8.7%) were applicable in the training period, but none were applicable in the rest period.

**Table 2 Single correlations between blood cell parameters, iron metabolism-related parameters, physical characteristics, and dietary factors and serum 25(OH)D.**

| Variable | Training | | Resting | |
|---|---|---|---|---|
| | r | p | r | p |
| BMI | 0.027 | 0.902 | −0.190 | 0.386 |
| Lean body mass | −0.162 | 0.461 | −0.085 | 0.700 |
| Skeletal muscle | −0.166 | 0.449 | −0.101 | 0.647 |
| Fat mass | 0.178 | 0.415 | −0.166 | 0.450 |
| Energy consumption | 0.114 | 0.606 | −0.145 | 0.508 |
| Energy intake | −0.074 | 0.737 | 0.346 | 0.106 |
| Energy intake/consumption | −0.200 | 0.361 | 0.305 | 0.157 |
| Dietary protein/BW | −0.314 | 0.145 | 0.317 | 0.140 |
| Dietary fat | 0.161 | 0.464 | 0.470* | 0.024 |
| Dietary carbohydrate | −0.040 | 0.858 | −0.381 | 0.073 |
| Dietary vitamin D | −0.091 | 0.680 | 0.432* | 0.040 |
| Dietary Fe | −0.039 | 0.859 | 0.246 | 0.258 |
| Red blood cell count | 0.152 | 0.488 | 0.006 | 0.977 |
| Hemoglobin | 0.269 | 0.214 | −0.129 | 0.558 |
| Hematocrit | 0.264 | 0.224 | 0.029 | 0.895 |
| Reticulocytes | 0.312 | 0.147 | −0.167 | 0.447 |
| Serum iron | 0.447* | 0.032 | −0.200 | 0.360 |
| Serum ferritin | 0.520* | 0.011 | 0.285 | 0.188 |
| TIBC | −0.290 | 0.180 | −0.151 | 0.492 |
| Serum transferrin saturation | 0.554* | 0.006 | −0.162 | 0.461 |
| Hepcidin-25 | 0.368 | 0.084 | 0.259 | 0.232 |
| Serum erythropoietin | −0.047 | 0.832 | −0.135 | 0.540 |

**Note:**
Spearman's rank correlation coefficient. *$p < 0.05$. BMI, body mass index; BW, body weight; TIBC, total iron-binding capacity.

## Correlations between serum 25(OH)D and iron metabolism factors during the training period

Correlations between blood cell parameters, iron metabolism-related parameters, physical characteristics, dietary factors, and serum 25(OH)D are shown in Table 2. Significant positive correlations were found between the lipid energy ratio (r = 0.470, $p$ = 0.024) and dietary vitamin D intake (r = 0.432, $p$ = 0.040) during the rest period. However, no significant correlations were observed with blood cells or iron metabolism-related items. In contrast, significant positive correlations were observed between serum 25(OH)D levels and serum iron (r = 0.447, $p$ = 0.032), ferritin (r = 0.520, $p$ = 0.011), and transferrin saturation (r = 0.554, $p$ = 0.006) levels during the training phase. Therefore, multiple regression analysis was performed to determine which factors were more strongly associated with serum 25(OH)D levels. Multiple regression analysis was conducted using the forced entry method with serum iron and ferritin as bivariate independent variables

**Table 3 Multiple regression analysis with serum 25(OH)D as the dependent variable during the training period.**

| Variable | Beta | Standardized coefficient Beta | p | 95% Confidence interval | VIF |
|---|---|---|---|---|---|
| Ferritin | 0.033 | 0.454 | 0.037 | [0.004–0.120] | 1.269 |
| Serum iron | 0.062 | 0.221 | 0.290 | [−0.030 to 0.095] | 1.269 |

**Note:**
Force entry method. $p = 0.014$, Adjusted $R^2 = 0.282$.

and 25(OH)D as the dependent variable. The results in Table 3 show that the association between serum 25(OH)D and ferritin was stronger and more significant than serum iron.

## DISCUSSION

This study aimed to clarify the changes in biological iron metabolism due to continuous training in young athletes by comparing the status of the same participants with and without training and examining the effect of vitamin D on these changes. The results showed that hemolysis and erythropoiesis were enhanced during the training phase, and serum levels of hepcidin, a regulator of iron metabolism, were altered. The correlation between serum 25(OH)D levels and iron metabolism factors was observed only in the training phase, with ferritin showing the strongest and most significant relationship. These results are an indication that there may be specific mechanisms of iron metabolism that are unique to continuous exercise loading. Although many studies have observed changes in hepcidin secretion during exercise, the originality of this study is that the same participants were used to observe changes in iron metabolism *in vivo* with and without continuous high-intensity and prolonged training. The finding that long-term training alters hepcidin secretion by factors other than IL-6 secretion and the finding of a relationship between vitamin D and biological iron *in vivo* during the training period are also new findings. In the following section, based on the present results, a discussion of how continuous training affects blood cell characteristics and iron metabolism and the involvement of vitamin D is presented.

First, this study focusses on hepcidin, a regulator of iron metabolism. Serum hepcidin-25 levels were significantly lower during the training period than during the rest period. Hepcidin is a regulator of iron metabolism released from the liver, and the suppression of hepcidin expression increases iron absorption from the gastrointestinal tract and tissue release. Factors that regulate hepcidin expression include (1) iron stores in the body, (2) acute or chronic inflammation, and (3) increased red blood cell production (*Peeling, 2010*).

Serum ferritin levels, an indicator of body iron stores, were not significantly different between the two periods, suggesting that their effects on hepcidin expression were small. In the present study, serum iron, transferrin saturation, and ferritin levels were not significantly different between the two periods. However, the median serum iron and transferrin saturation during the training period were lower than during the rest period, and the data distribution was wider. Lower hepcidin levels in the training phase may have increased iron absorption from the gastrointestinal tract and iron release from macrophages, thereby preventing iron deficiency. TIBC during the training period was

significantly lower than that during the rest period to lower serum iron and apo-transferrin levels, suggesting that transferrin expression does not increase because the organism does not recognize it as being iron-deficient.

Second, IL-6, one of the indicators of acute or chronic inflammation, depends on the signal transducer and activator of the transcription 3 (STAT3) signaling pathway to promote hepcidin expression. IL-6 is a cytokine produced by the liver, muscle, and adipocytes. It can be muscle-derived, which increases immediately after exercise, or immune cell-derived, which increases several hours after exercise in response to cell injury or infection (*Fischer, 2006*). IL-6 levels increase immediately after exercise and return to baseline 3–6 h later (*Hennigar, McClung & Pasiakos, 2017*). In the present study, IL-6 was detectable in both the training and rest periods in three of the 23 participants, and the average of these three participants was similar in both periods. Thus, the promotion of hepcidin expression by IL-6 production may not occur during continuous training.

Third, the RBC, Hb, and Ht levels were significantly lower during the training period than during the rest period. Ferritin level, serum iron level, and transferrin saturation, which are indicators of body iron content, were not significantly different between the two groups, suggesting that iron deficiency did not occur. Although not shown in the results, haptoglobin levels in all participants were lower during the training period than during the rest period for all types, suggesting that hemolysis may have occurred due to continuous training over the 2 months. The reticulocyte count and serum EPO concentration were higher during the training phase than during the resting phase. Reticular erythrocytes are an indicator of increased erythropoiesis in the bone marrow immediately after the maturation and denucleation of erythroblasts (*Stevens-Hernandez & Bruce, 2022*). EPO, a regulator of erythropoiesis, enhances hematopoiesis by acting on erythroid progenitor cells and erythroblasts before Hb synthesis (*Heeschen et al., 2003*). Therefore, it can be inferred that hemolysis and erythropoiesis are converted into hyperproduction during the training phase, which may suppress hepcidin expression. Thus, continuous training suppressed hepcidin expression, which may be partly due to the increased erythropoiesis caused by hemolysis.

Next, to investigate the effect of vitamin D on exercise, correlations were observed between serum 25(OH)D levels and blood characteristics, physical characteristics, iron metabolism-related items, and nutrient intake during exercise and rest. Because the population in this study was a mixture of normally and non-normally distributed variables, correlation analysis was performed using Spearman's rank correlation coefficient. Significant positive correlations were observed between serum iron, ferritin, and transferrin saturation levels during the training period. However, no correlation was observed between these items during the rest period, suggesting that this phenomenon occurred only during the training period. Incidentally, no significant associations were also found when correlation analysis was performed without grouping. This also supports the idea that exercise loading may produce a significant association between vitamin D and biological iron.

Furthermore, multiple regression analysis with serum 25(OH)D as the dependent variable and serum iron and ferritin levels as the independent variables during the training

period showed that ferritin levels were significantly associated with serum 25(OH)D levels. The relationship between serum 25(OH)D and serum ferritin levels is debatable with only a few reports. For example, a significant positive correlation was observed between serum 25(OH)D and ferritin in adolescents (*Andıran et al., 2012*). Serum 25(OH)D levels are inversely correlated with serum ferritin levels in males but positively correlated in premenopausal women (*Seong et al., 2017*); however, consistent results have not been obtained (*Munasinghe et al., 2019*). Only a few reports have mentioned the effects of exercise load. *In vitro* validation has shown that iron deficiency reduces the activity of heme-containing vitamin D-activating enzymes, such as 25- and 1α-hydroxylase (*Katsumata et al., 2016*), and induces vitamin D deficiency by promoting the transcription of fibroblast growth factor 23 (FGF23), which inhibits vitamin D 1α-hydroxylation (*Clinkenbeard et al., 2014*). Iron deficiency has also been recently reported to affect hydroxylase activity by decreasing iron levels in the liver and kidneys and reducing serum 25(OH)D3 and 1,25(OH)2D levels (*Qiu et al., 2022*). However, some studies have shown that high doses of vitamin $D_3$ ($VD_3$) reduce hepcidin levels (*Munasinghe et al., 2019*). *In vitro* experiments have reported that the 1,25(OH)2D -vitamin D receptor complex binds to the vitamin D response element (VDRE) on the hepcidin gene (HAMP) and represses its transcription (*Smith & Tangpricha, 2015*), suggesting a negative feedback-like regulatory mechanism by which VD3 increases body iron *via* hepcidin repression. Furthermore, a notable phenomenon in this study was that serum 25(OH)D was significantly positively correlated with dietary vitamin D intake and the lipid energy ratio during the resting period; however, this correlation was lost during the training period. These findings suggest a change in $VD_3$ consumption and metabolism during exercise. This may be due to the effects of FGF23 and the parathyroid hormone PTH. FGF23 is reportedly stimulated by exercise for secretion (*Li et al., 2016*). FGF23 has been implicated in vitamin D activity and negatively regulates erythropoiesis and iron metabolism (*Coe et al., 2014*). High concentrations of FGF23 have been reported to suppress 1α-hydroxylase activity, resulting in lower 1,25(OH)2D concentrations (*Clinkenbeard et al., 2014*). PTH is known to increase in an exercise intensity- and time-dependent manner (*Scott et al., 2011*), and its concentration is inversely correlated with the serum 25(OH)D concentration (*Lips et al., 2001*).

A limitation of this study is that it is an observational study with a small number of participants. Moreover, as the dietary and physical activity surveys are self-reported responses by the participants, the possibility of psychological selection bias and differences from the actual content cannot be ruled out. The influence of the food environment, menstruation, physical development rate, and food preferences was not investigated; therefore, their relevance to anemia is unknown. As the present study did not examine different sexes and age groups, further studies are needed. However, the strength of this study is that the metabolic changes observed with and without exercise in the same participants are unique and valuable data. The participants are of the same age group and train together daily on the same training menu, so it can be inferred that the bias in the amount of exercise load is as small as possible. The data examining the relationship between dietary vitamin D intake and 25(OH)D in a Japanese sample are valuable, and the

fact that this relationship changes during exercise and that there are many data on vitamin D and bone and muscle formation, but the relationship with biological iron is a new finding and strengthens the significance of this study.

## CONCLUSIONS

The present study examined the changes in biological iron metabolism during continuous training load and the effects of vitamin D on these changes, suggesting that continuous training enhances hemolysis and erythropoiesis and may be a factor in the suppression of hepcidin expression. Multiple regression analysis revealed a significant relationship between serum 25(OH)D and ferritin levels only during the training period, suggesting that the relationship between 25(OH)D and iron levels may be closely related to metabolic changes in the body due to exercise load. In the future, we would like to focus on the relationship between iron and vitamin D during exercise, including the observation of iron metabolic changes with vitamin D supplementation, to clarify the role of iron and vitamin D in maintaining athletes' health. In recent years, cases of iron overload in the body induced by iron injections or administration in the hope of hemopoiesis in athletes have come to light, but if the contribution of dietary vitamin D to the specific iron metabolism mechanisms produced by continuous exercise loading can be made explicit, it could help to inform dietary strategies for the prevention of iron deficiency in athletes.

## ACKNOWLEDGEMENTS

We would like to thank Hiroshi Kawabata, M.D., Ph.D., Kyoto National Hospital Organization, Kyoto Medical Center, for supporting this research. We would like to thank the director of the high school dance club and the members of the club for their cooperation in conducting this study.

### Funding

This work was supported by the JSPS KAKENHI Grant-in-Aid for Scientific Research (C) (grant number 19K11695), Japan Society for the Promotion of Science. The funders had no role in study design, data collection and analysis, decision to publish, or preparation of the manuscript.

### Grant Disclosures

The following grant information was disclosed by the authors:
JSPS KAKENHI Grant-in-Aid for Scientific Research (C): 19K11695.

### Competing Interests

The authors declare that they have no competing interests.

## Author Contributions

- Yukiko Kobayashi conceived and designed the experiments, performed the experiments, analyzed the data, prepared figures and/or tables, authored or reviewed drafts of the article, and approved the final draft.
- Rikako Taniguchi conceived and designed the experiments, performed the experiments, analyzed the data, prepared figures and/or tables, and approved the final draft.
- Emiko Shirasaki conceived and designed the experiments, performed the experiments, analyzed the data, prepared figures and/or tables, and approved the final draft.
- Yuko Segawa Yoshimoto performed the experiments, analyzed the data, authored or reviewed drafts of the article, and approved the final draft.
- Wataru Aoi conceived and designed the experiments, authored or reviewed drafts of the article, and approved the final draft.
- Masashi Kuwahata conceived and designed the experiments, authored or reviewed drafts of the article, and approved the final draft.

## Human Ethics

The following information was supplied relating to ethical approvals (*i.e.*, approving body and any reference numbers):

This study was conducted with the approval of the Kyoto Prefectural University Ethics Committee (No. 207).

## Data Availability

The raw data is available in the Supplemental Files.

## Supplemental Information

Supplemental information for this article can be found online at http://dx.doi.org/10.7717/peerj.17566#supplemental-information.

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
