# Peer review of "Continuous training in young athletes decreases hepcidin secretion and is positively correlated with serum 25(OH)D and ferritin"

_PeerJ, doi:10.7717/peerj.17566_

## Round 0.1 · original submission · Major Revisions

The feedback provided by the reviewers is quite consistent and suggests that the manuscript faces some fundamental challenges.

Within the Introduction, the manuscript falls short in clearly delineating the study's objectives, its significance within the field, and the underlying hypothesis. Hypotheses play a pivotal role in hypothesis-testing research, yet regrettably, they are omitted in this manuscript.

Furthermore, there is a notable absence of any mention regarding Statistical Analysis within the Materials and Methods section. In hypothesis-testing research, it is imperative to outline how the sample was collected and analyzed in accordance with the stated hypothesis. This critical aspect is unfortunately lacking in this manuscript.

Table 2 presents the results of the correlation analysis, yet it remains unclear which hypothesis guided this particular analysis. Additionally, there are concerns regarding the potential for Type-I errors, especially given the extensive number of analyses conducted. Strategies for mitigating such errors should be elucidated within the manuscript.

By addressing these points, the manuscript can significantly enhance its clarity and adherence to established research norms.

·

Basic reporting

The study needs some important revisions regarding its sample and methodology

Experimental design

Power analysis details should be given. How was the study sample determined?
What was the power realized in the work? Which program was used for power analysis?

There is no detailed information about statistical analysis.

On what basis were the tests decided and how was the suitability of the data for normal distribution or homogeneity of variances examined?

Validity of the findings

p values should be given in italics.

Give correlation results according to APA style.

Improve the quality of figures.

Additional comments

Additionally, it is important to emphasize the innovative aspect of the study in more detail in the introduction section.

·

Basic reporting

Thank you for your submitted manuscript entitled, “Continuous training in young athletes decreases hepcidin secretion and is positively correlated with serum 25(OH)D and ferritin’’. The area of the research is interesting; however, it needs a few amendments. Overall, the paper is well-written. The manuscript is well-structured, and it is easy to follow the sections.

The Abstract - It´s not clear the aim of the study.
The key words should be different from the words in the title.

The introduction provides quite a proper background of the topic, but the rationale for examining this problem should be mentioned more clearly in this section. Why did the authors choose to examine it? Why didn’t the authors choose to study some other aspects?
The article innovation should be presented in the last part of the Introduction. Describe what the research gap of the paper is and what is new. Please describe the links between the research gap and the goal of the article.

The quality of the images is good enough, but I don’t know if the reviewing version has lower resolution than the final version. If not, images should have better resolution in its final size.
It seems that the English is clear, but research articles usually do not use the word "we" and regularly use passive verbs (lines 22, 84, 188, 207, 210, 249).

Experimental design

The experimental design meets the scope of the journal, and the methods are described detailed enough.
Were the activities tested under the same conditions?
How was the statistical analysis conducted? It is suggested to create a subsection.
How was the normality of the data checked? Was the sample power calculated?
The three tables and Figure 1 illustrate the relationships presented in the study.

Validity of the findings

Most of the results are quite interesting and are well discussed.
In the Discussion it would be better to have seen more use of terms like 'originality' and 'significance' in the first paragraph. Identify what is new in this study that may benefit readers or how it may advance existing knowledge or create new knowledge on this subject. There should be a clear conclusion on why the research findings are significant (the limitations are well presented but the strengths somehow are overlooked).
The authors should introduced a sentence with future research proposals.

---

## Round 0.2 · Minor Revisions

Before considering publication, the following points should be addressed in your manuscript:

1. Please clarify the methodology used to calculate the sample size.

2. Please provide the results of the power analysis.

3. Please justify the necessity of conducting Spearman's rank correlation test.

4. After checking your data, I assume that the results of the statistical analysis without grouping would differ from the results you have presented in this manuscript. If so, please compare your results with those obtained when grouping was performed and discuss the possible reasons for the differences.

·

Basic reporting

Some significant statistical revisions are necessary.

Experimental design

When you use the Wilcoxon signed-rank sum test, you cannot use mean and standard deviation as descriptive statistics. Instead, you should use median and interquartile range (IQR).

Use separate markers for Paired t-test or Wilcoxon signed-rank test in tables, maybe first test *, second test ** (The same situation is valid for Figure 1.)


Descriptive statistics in Figure 1 should be reviewed and expressed as median and IQR in the non-parametric case.

Validity of the findings

no comment.

Additional comments

Once the specified items are completed, the article can be accepted.

·

Basic reporting

Thank you for providing this comprehensive work.
The authors have presented an improved version of the manuscript.

The introduction provides a proper background of the topic.
The sections have been improved. Relevant results are well-organised.
The quality of the images is good enough.

Experimental design

The experimental design meets the scope of the journal, and it is relevant to the community.
Methods are described detailed enough.

Validity of the findings

The results and the conclusions are quite interesting and well-discussed. All data are provided.


The authors have adequately addressed all my comments. I have no further suggestions.

Additional comments

The authors have adequately addressed all my comments. I have no further suggestions.

---

## Round 0.3 · accepted · Accept

I have read the current version of your manuscript. I have concluded that the manuscript has been appropriately revised and is of great interest to readers.